# The FDA-Approved Anti-Asthma Medicine Ciclesonide Inhibits Lung Cancer Stem Cells through Hedgehog Signaling-Mediated SOX2 Regulation

**DOI:** 10.3390/ijms21031014

**Published:** 2020-02-04

**Authors:** Hack Sun Choi, Su-Lim Kim, Ji-Hyang Kim, Dong-Sun Lee

**Affiliations:** 1Subtropical/Tropical Organism Gene Bank, Jeju National University, Jeju 63243, Korea; 2School of Biomaterials Science and Technology, College of Applied Life Science, Jeju National University, Jeju 63243, Korea; 3Interdisciplinary Graduate Program in Advanced Convergence Technology & Science, Jeju National University, Jeju 63243, Korea; 4Faculty of Biotechnology, College of Applied Life Sciences, Jeju National University, SARI, Jeju 63243, Korea; 5Practical Translational Research Center, Jeju National University, Jeju 63243, Korea

**Keywords:** lung cancer stem cells, ciclesonide, Hedgehog signaling, GL1, GL2, SMO

## Abstract

Ciclesonide is an FDA-approved glucocorticoid (GC) used to treat asthma and allergic rhinitis. However, its effects on cancer and cancer stem cells (CSCs) are unknown. Our study focuses on investigating the inhibitory effect of ciclesonide on lung cancer and CSCs and its underlying mechanism. In this study, we showed that ciclesonide inhibits the proliferation of lung cancer cells and the growth of CSCs. Similar glucocorticoids, such as dexamethasone and prednisone, do not inhibit CSC formation. We show that ciclesonide is important for CSC formation through the Hedgehog signaling pathway. Ciclesonide reduces the protein levels of GL1, GL2, and Smoothened (SMO), and a small interfering RNA (siRNA) targeting SMO inhibits tumorsphere formation. Additionally, ciclesonide reduces the transcript and protein levels of SOX2, and an siRNA targeting SOX2 inhibits tumorsphere formation. To regulate breast CSC formation, ciclesonide regulates GL1, GL2, SMO, and SOX2. Our results unveil a novel mechanism involving Hedgehog signaling and SOX2 regulated by ciclesonide in lung CSCs, and also open up the possibility of targeting Hedgehog signaling and SOX2 to prevent lung CSC formation.

## 1. Introduction

Lung cancer, known as lung carcinoma, is a common cancer and is known to cause uncontrolled cell growth in lung tissue. Lung cancer is a major cause of cancer death, accounting for 20% of all cancer-related deaths [1]. Chemotherapy improves survival rate to a limited extent, but the overall survival rate is low due to the recurrence of aggressive tumors [2]. Lung cancer is divided into two groups based on pathological properties. These two types are non-small cell lung cancer (NSCLC) and small cell lung cancer (SCLC). NSCLC accounts for 85% of all lung cancers, and the survival rate is low [3].

Cancer stem cells (CSCs) were first identified at Dr. John Dick’s laboratory [4], and the concept of CSCs has become an interesting field in cancer research. CSCs represent a small portion of the total cancer cell population and have strong tumorigenic properties [5]. The characteristics of CSCs are self-renewal, differentiation, tumorigenicity, and resistance to chemotherapy [6,7]. CSC markers in lung cancer include the metabolic marker aldehyde dehydrogenase isoform 1 (ALDH1) and the surface markers CD133, CD44, and CD166 [8]. Novel therapeutic methods targeting CSCs can improve long-term clinical outcomes [1].

Hedgehog signaling is activated during embryogenesis and development, and reactivated in many solid tumors [9,10,11]. The proteins involved in Hedgehog signaling are divided into Sonic Hedgehog (Shh), Indian Hedgehog (Ihh), and Desert Hedgehog (Dhh) according to species. In humans, Sonic Hedgehog is the main protein. The Hedgehog signaling pathway consists of three proteins, the GPCR-like protein Smoothened (SMO), the canonical receptor Patched (PTCH1), and the GLI1/2/3 proteins. Hedgehog signaling is essential for self-renewal, cell fate determination, and CSC formation [11,12]. Aberrant Hedgehog signaling is associated with the progression of several cancers, tumorigenesis, and CSC formation [12]. Additionally, the Hedgehog signaling pathway is involved in chemo-resistance, relapse, and metastasis of lung cancer [12].

SOX2 is a key transcription factor for maintaining self-renewal, reprogramming of somatic cells, and pluripotency of undifferentiated embryonic stem cells [13,14,15]. The SOX2 protein is involved in several cancers, including melanoma, lung cancer, breast cancer, ovarian cancer, and pancreatic cancer [2,16,17,18,19], and maintains CSCs in skin, bladder, and colorectal cancers [20]. Other groups previously showed that Hedgehog-GLI signaling is involved in cancer growth, tumorigenicity, and stemness [16,21,22]. Through Hedgehog-GLI signaling, GLI regulates the *SOX2* gene, and GLI-mediated regulation of SOX2 induces self-renewal of melanoma and lung CSCs [2,16].

The anti-asthma medicine ciclesonide is a glucocorticoid (GC) used to treat asthma and allergic rhinitis. Ciclesonide, a new inhaled corticosteroid, is effective as a once-daily controller therapy for pediatric asthma and reduces airway inflammation through a single daily administration [23]. Ciclesonide has strong anti-inflammatory activity in vitro and in vivo. The relative binding affinities of ciclesonide for the rat GR were higher than those of dexamethasone [24]. We demonstrated that ciclesonide had anti-proliferative properties against lung cancer and inhibited lung CSC formation through suppression of Hedgehog signaling and SOX2. Hedgehog signaling and SOX2 are potential therapeutic targets for CSCs in lung cancer.

## 2. Results

### 2.1. Ciclesonide Inhibits Proliferation and Induces Apoptosis in A549 Lung Cancer Cells

We assessed the effect of ciclesonide on the growth of A549 human lung cancer. Ciclesonide showed anti-proliferative effects (Figure 1A). After treatment of lung cancer with ciclesonide, the formation of apoptotic bodies was induced (Figure 1B). Ciclesonide induced apoptosis of lung cancer (Figure 1C). Ciclesonide increased caspase 3/7 activity in A549 cells (Figure 1D). Ciclesonide inhibited migration and colony formation of lung cancer (Figure 1E,F). We showed that ciclesonide effectively suppressed lung cancer cell growth, apoptosis, cell migration, and colony formation.

### 2.2. Ciclesonide Blocks Tumor Growth

Since ciclesonide inhibited the proliferation of lung cancer, we tested whether ciclesonide reduces tumor growth in a mouse model. Nude mice in the control and ciclesonide-treated groups had similar body weights (Figure 2A). The tumor weights of ciclesonide-treated mice were lower than the control mice (Figure 2B). The tumor volumes of the ciclesonide-treated mice were smaller than the control mice (Figure 2C). We showed that ciclesonide reduced tumor growth in the mouse model.

### 2.3. Effect of Ciclesonide, Prednisone, and Dexamethasone on Lung CSCs

To examine whether ciclesonide inhibits tumorsphere formation in lung cancer, we cultured tumorspheres derived from A549 cells with different concentrations of ciclesonide. Ciclesonide and isobutyryl ciclesonide inhibited lung tumorsphere formation (Figure 3A,B). We evaluated cell proliferation and tumorsphere formation using the most popular steroid hormones, prednisone and dexamethasone. These two steroids did not affect cell growth or tumorsphere formation (Figure 3C,D). We checked the ALDH1 level in lung cancer cells because it is a marker of lung CSCs. We examined the effect of ciclesonide on ALDH1-positive lung cancer cells. Ciclesonide decreased the ALDH1-positive cell fraction from 5.0% to 2.5% (Figure 4). Our results showed that ciclesonide specifically suppresses tumorsphere formation.

### 2.4. Ciclesonide Inhibits Tumorsphere Formation through Inhibition of Hedgehog Signaling

To determine the cellular mechanism of ciclesonide in tumorigenesis, we examined the levels of GLI transcripts and proteins in the tumorsphere under ciclesonide treatment because ciclesonide and budesonide inhibit the Hedgehog signaling pathway [25]. Our results showed that transcripts of GLI1 and GLI2 but not GLI3 were reduced significantly under ciclesonide treatment (Figure 5A). To provide a biological meaning of the GLI1 and GLI2 genes in lung cancer, we used the Xena browser analysis for comparison of normal and tumor GLI gene transcripts. The Xena browser analysis indicated a higher gene expression of GLI1 and GLI2 in primary tumor tissue compared to normal tissue (Appendix A) (GLI1; *n* = 786, *p*-value = 2.894 × 10^−21^, GLI1; *n* = 786, *p*-value = 6.001 × 10^−33^). Ciclesonide reduced GLI1, GLI2, and SMO protein levels (Figure 5A). To check SMO function in tumorsphere formation, we examined the tumorsphere formation ability of cells subjected to small interfering RNA (siRNA) silencing of SMO. Lung cancer cells transfected with SMO-specific siRNA showed a 40% reduction in tumorsphere formation (Figure 5B). Our observations suggest that SMO and Hedgehog signaling are important for tumorsphere formation in lung cancer. To confirm the specificity of the ciclesonide effect, we checked the Hippo-YAP signaling with ciclesonide present in A549 lung cancer cells. Ciclesonide did not change YAP or pYAP in lung cancer (Appendix A).

### 2.5. Ciclesonide Inhibits Tumorsphere Formation through GLI-Mediated SOX2 Regulation

Since GLI1-mediated regulation of SOX2 regulates the self-renewal of lung and melanoma CSCs [2,16], we examined SOX2 levels using ciclesonide. Ciclesonide reduced the transcript and protein levels of SOX2 (Figure 6A). In order to confirm the regulation of the *SOX2* gene by Hedgehog signaling, we performed knock-down of the SMO gene using siRNA of the SMO gene and checked SOX2 transcripts. SMO regulated *SOX2* gene expression (Figure 6A). To investigate SOX2 function in tumorsphere formation, we examined the tumorsphere formation ability of cells via siRNA silencing of SOX2. Lung cancer cells transfected with SOX2-specific siRNASOX showed a 70% reduction in tumorsphere formation (Figure 6B). Our observations suggest that ciclesonide reduced SOX2 and Hedgehog signaling, which is important for tumorsphere formation in lung cancer.

### 2.6. Ciclesonide Inhibits Gene Expression of Cancer Stem Cell Markers and Growth of Tumorspheres

To determine whether ciclesonide inhibits lung CSC-specific genes, we checked the transcript levels of CSC-specific genes. Ciclesonide decreased the transcription levels of the specific genes SOX2, Nanog, c-Myc, and Snail in lung CSCs (Figure 7A). To confirm that ciclesonide reduced tumorsphere growth, we added ciclesonide to tumorsphere medium and cultured the cancer cells derived from tumorspheres. Ciclesonide induced cell death of the tumorspheres (Figure 7B). We showed that ciclesonide decreased the growth of tumorspheres and that ciclesonide suppressed lung CSC formation through dysregulation of the Hedgehog/SOX2 signaling axis.

## 3. Discussion

Lung cancer is a life-threatening disease, and its death rate is higher than that of other cancers. Lung cancer accounts for 20% of cancer-related deaths [26]. Of all lung cancers, 85% are NSCLCs. Many scientists believe that CSCs or cancer-initiating cells are a subpopulation of cancer cells that confer aggressiveness and chemo-resistance to cancer cells. CSCs have the capacity to self-renew, initiate tumors, and undergo multiple differentiations. Therefore, the properties of lung CSCs have become targets of lung cancer therapies.

In this report, we showed that ciclesonide suppresses the proliferation of lung cancer cells and the growth of CSCs. Similar glucocorticoids, such as dexamethasone and prednisone, do not inhibit CSC formation. In the hospital, glucocorticoids such as dexamethasone (DEX) have been administered to cancer patients to minimize chemotherapy, reduce nausea, and protect healthy tissue [27]. Clinical evidence has shown GR-induced chemotherapy resistance and poor prognosis in cancers [28,29]. Another GC, budesonide, and ciclesonide are distinguished by bulky hydrophobic groups at positions 16 and 17 [25]. Ciclesonide, dexamethasone and prednisone showed different effects on lung CSC formation based on their different structures. Prednisone promoted SMO accumulation at cilia and induced Hedgehog stimulation, but ciclesonide and budesonide inhibited SMO ciliary localization and signaling activity [25]. Cyclopamine, a potent Hedgehog signaling antagonist, inhibits breast cancer growth independent of SMO [30].

We showed that ciclesonide reduced transcripts of GLI1 and GLI2 and played an important role in lung CSC formation by using an siRNA to target SMO. Our data showed that ciclesonide inhibits lung CSCs through regulation of Hedgehog signaling. GCs induce chemo-resistance and tumor relapse by promoting the unexpected growth of CSCs that are resistant to therapy and highly metastatic [31,32,33]. Ciclesonide has a different function from that of typical GCs due to its different structure.

Our data showed that anti-proliferative concentrations of ciclesonide in mouse and cell experiments are 600 and 300 mg/60 kg, respectively. The concentration of ciclesonide used in our experiments is very high (ciclesonide drug dosage; 200 μg/1 day and 73 mg/1 year) and may induce cytotoxicity. It is very hard to use ciclesonide as a cancer drug. In order to overcome this problem, we have two options; one is the development of a ciclesonide analog that has strong anti-cancer activity, another is ciclesonide co-medication in cancer patients undergoing chemotherapy. 

CSCs are responsible for drug resistance, recurrence, and metastasis, which are major causes of cancer mortality [34]. Aberrant Hedgehog signaling induced progression and tumorigenesis, including CSC maintenance, in several cancers [11,12]. SOX2 regulated the self-renewal of human melanoma-initiating cells [16]. Our data showed that ciclesonide reduced Hedgehog signaling and SOX2 expression. Ciclesonide-mediated regulation of the Hedgehog signaling/SOX2 pathway regulates lung CSC formation.

Ciclesonide, a new inhaled corticosteroid, reduces airway inflammation through a single daily administration and controls asthma in atopic children [23]. Ciclesonide and budesonide inhibit the Hedgehog signaling pathway [25]. Ciclesonide inhibits the function of Hedgehog signaling and the SOX2 protein. Hedgehog signaling and the SOX2 protein regulate lung CSC formation. Our results unveil a novel mechanism involving Hedgehog signaling and SOX2 protein expression induced by ciclesonide in lung CSCs, and also open up the possibility of targeting Hedgehog signaling to prevent CSC formation. Ciclesonide may show potential for applications in anticancer therapy and inflammation.

## 4. Materials and Methods

### 4.1. Culture and Tumorsphere Formation Assay

A549 human lung cancer cells were obtained from the American Type Culture Collection (Rockville, MD, USA) and maintained in RPMI medium with 10% fetal bovine serum (HyClone, Thermo Fisher Scientific, CA, USA) and 1% penicillin/streptomycin (HyClone, Thermo Fisher Scientific, CA, USA). For lung tumorsphere formation, 5 x 10^4^ human lung cancer cells were cultured in an ultralow-adherence plate with Cancer Stem Premium Medium (ProMab Biotechnologies Inc., Richmond, CA, USA). All cells were maintained in a humidified 5% CO_2_ incubator at 37 °C for 7 days. The number of tumorspheres was estimated by using the NICE program [35]. Formation of tumorspheres was estimated by determining tumorsphere formation efficiency (TFE) (%) [36].

### 4.2. Antibodies and siRNAs

Anti-GLI1, anti-GLI2, anti-SMO, and anti-SOX2 antibodies were purchased from Cell Signaling Technology (Danvers, MA, USA). Anti-β-actin antibody was obtained from Santa Cruz Biotechnology. Anti-CD44 FITC and anti-CD24 PE antibodies were obtained from BD Pharmingen (BD, San Jose, CA, USA). Human SMO- and SOX2-specific siRNAs were obtained from Bioneer Corp. (Daejeon, Korea).

### 4.3. Cell Proliferation

We followed a previously described method [37]. A549 lung cancer cells were cultured in a 96-well plate with ciclesonide. The proliferation assay was assessed using a CellTiter 96® Aqueous One Solution cell kit (Promega, Madison, WI, USA), and the OD_490_ was measured using a microplate reader (SpectraMax, San Jose, CA, USA).

### 4.4. Colony Formation and Migration Assay

For the colony formation assay, A549 cells were incubated at 1000 cells/well with ciclesonide for 7 days in RPMI/10% FBS. The colonies were cultured and counted. For the migration assay, the cells were cultured in a 24-well plate, and a scratch was made by using a pipette tip. After washing with RPMI/10% FBS, the lung cancer cells were treated with ciclesonide. We used a previously described method [38].

### 4.5. Annexin V/PI Assay and Analysis of Apoptosis

Lung cancer cells were cultured in 6-well plates with ciclesonide (20 μM). Apoptotic cells were detected by Annexin V/PI staining according to the manufacturer’s instructions (BD, San Jose, CA, USA). The samples were analyzed by Accuri C6 (BD, San Jose, CA, USA). Cancer cells were treated with 20 μM ciclesonide for 1 day, and then the cells were incubated with Hoechst 33258 (10 mg/mL) solution for 20 min at 37 °C. The cells were observed using a fluorescence microscope (Lionheart FX, Biotek, Winooski, VT, USA).

### 4.6. Hoechst Staining and ALDEFLUOR Assay

For Hoechst 33258 staining, A549 lung cancer cells were treated with 30 μM ciclesonide for 1 day, and then the cells were incubated with Hoechst 33258 (10 mg/mL) solution for 30 min at 37 °C. The cells were observed using a fluorescence microscope (Lionheart FX, Biotek, Winooski, VT, USA). The aldehyde dehydrogenase activity was assayed using an ALDEFUOR^TM^ assay kit (STEMCELL Technologies, Vancouver, BC, Canada). We used a previously described method [38]. Cells were cultured in ALDH assay buffer at 37 °C for 30 min. ALDH-positive cells were counted using an Accuri C6 (BD, San Jose, CA, USA).

### 4.7. Gene Expression Analysis

Total RNA from cancer cells was extracted and purified, and quantitative real-time RT-PCR was assayed using a real-time One-step RT-qPCR kit (Enzynomics, Daejeon, Korea). We used a previously described method [37]. The RT-qPCR primers are described in Appendix A.

### 4.8. Western Blot Analysis

Protein extracts were extracted from lung cancer cells and tumorspheres. After 12% SDS-PAGE, the proteins were transferred to a polyvinylidene fluoride (PVDF) membrane (EMD Millipore, Burlington, MA, USA). The membranes were incubated in Odyssey blocking buffer for 1 h and then incubated with primary antibodies. Anti-SMO, anti-GLI1, anti-GLI2, anti-Sox2, and anti-β-actin antibodies were used. After washing, the membranes were reacted with IRDye-conjugated secondary antibodies. Images were captured using an ODYSSEY CLx system (LI-COR, Lincoln, NE, USA).

### 4.9. Caspase-3/7 Assay

We used a previously described method [39]. The lung cancer cells were incubated with ciclesonide (40 and 80 μM). Caspase-3/7 activity was determined according to the manufacturer’s instructions using the Caspase-Glo 3/7 kit (Promega, Madison, WI, USA). One hundred microliters of Caspase-Glo 3/7 reagent was added to 96-well plates and incubated, and the activity was measured using a GloMax® Explorer plate-reading luminometer (Promega, Madison, WI, USA).

### 4.10. Small Interfering RNA (siRNA)

To examine the effect of SOX2 and SMO on the formation of tumorspheres, we transfected cancer cells with human SOX2 and SMO siRNA (Bioneer, Daejeon, South Korea). SOX2 siRNA (NM_003106.3) and SMO siRNA (NM_181651.1) were obtained from Bioneer Corp. (Daejeon Corp., South Korea). For the transfection, cancer cells were incubated and transfected using Lipofectamine 3000 (Invitrogen, Carlsbad, CA, USA) according to the manufacturer’s protocol. The protein levels of SOX2 and SMO were investigated via Western blot analysis.

### 4.11. Xenograft Transplantation

Twelve male nude mice were injected with two million A549 cells with/without ciclesonide (10 mg/kg). Tumor volumes were estimated for 55 days using the formula (width^2^ × length)/2. The experiments were performed as described previously [40]. Animal care and experiments were performed with protocols approved by the Institutional Animal Care and Use Committee (IACUC) of Jeju National University. Male nude mice (4 weeks old) were purchased from OrientBio (Seoul, South Korea) and kept in mouse facilities for 1 week.

### 4.12. Statistical Analysis

All data were computed with GraphPad Prism 5.0 software (GraphPad Prism Inc., San Diego, CA, USA). All data values are reported as the means ± standard deviations. Data were analyzed by using one-way ANOVA. Significant values were those with *p* < 0.05.

## 5. Conclusions

In this study, we showed that ciclesonide suppresses the proliferation of lung cancer cells and the growth of CSCs. Similar glucocorticoids, such as dexamethasone and prednisone, do not inhibit CSC formation. We show that ciclesonide is important for CSC formation through the Hedgehog signaling pathway. Ciclesonide reduced the protein levels of GL1, GL2, and SMO, and an siRNA targeting SMO inhibited tumorsphere formation. Ciclesonide reduced the transcript and protein levels of SOX2, and an siRNA targeting SOX2 inhibited tumorsphere formation. Ciclesonide-mediated regulation of GL1, GL2, SMO, SOX2, Hedgehog signaling, and the SOX2 pathway regulates breast CSC formation. Our results unveil a novel mechanism involving Hedgehog signaling and SOX2 expression induced by ciclesonide in lung CSCs, and also open up the possibility of targeting Hedgehog signaling and SOX2 to prevent lung CSC formation.

## Figures and Tables

**Figure 1 ijms-21-01014-f001:**
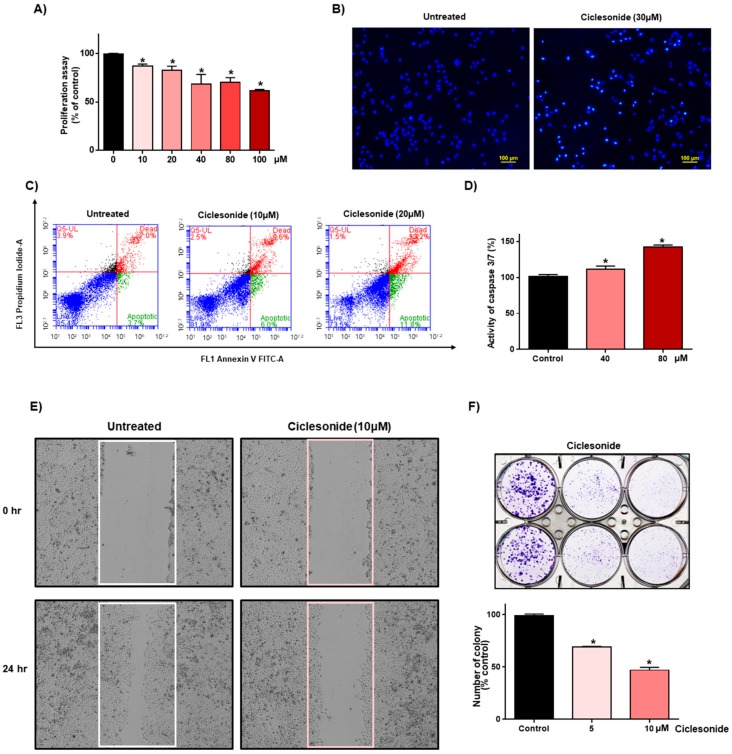
Ciclesonide reduces the proliferation of lung cancer. (**A**) A549 lung cancers were cultured in a 96-well plate with ciclesonide. The growth of cancer cells was assayed with an MTS reagent. (**B**) Ciclesonide (30 μM) induced apoptotic bodies stained with Hoechst 33342 dye (magnification, 40×). (**C**) Ciclesonide (10 and 20 μM) induced apoptosis of A549 cells. Apoptosis was assayed using Annexin V/PI staining. (**D**) Caspase 3/7 activity was measured by a caspase-Glo assay kit (Promega). Ciclesonide induced caspase 3/7 activity in A549 cells. (**E**) Ciclesonide (10 μM) inhibited cell migration, evaluated by a scratch assay. (**F**) The inhibitory effect of ciclesonide on colony formation of A549 cells. A total of 1 × 10^3^ cancer cells were incubated in 6-well plates containing 5 and 10 μM of ciclesonide. The data of triplicate experiments are represented as the mean ± SD; * *p* < 0.05, compared with control.

**Figure 2 ijms-21-01014-f002:**
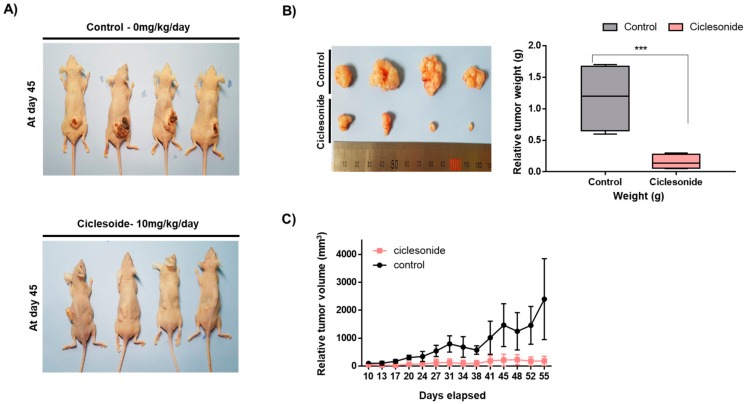
Ciclesonide reduces tumor growth in a mouse model. A total of 2 × 10^6^ lung cells were injected into nude mice subcutaneously. Inhibitory effect of ciclesonide on tumor growth in A549-bearing mice. (**A**) The concentration of ciclesonide is 10 mg/kg. After 55 days, images of the mice were captured. (**B**) Inhibitory effect of ciclesonide on tumor weight. The nude mice were sacrificed on day 55, and tumor weights were assayed. The data of triplicate experiments are shown as the mean ± SD. Compared with control, *** *p* < 0.05. (**C**) The tumor volumes of the mice over 55 days were comparable between the control and ciclesonide-treated mice. Tumor volumes were calculated as (width^2^xlength)/2. The ciclesonide-treated group exhibited a decrease in tumor volume.

**Figure 3 ijms-21-01014-f003:**
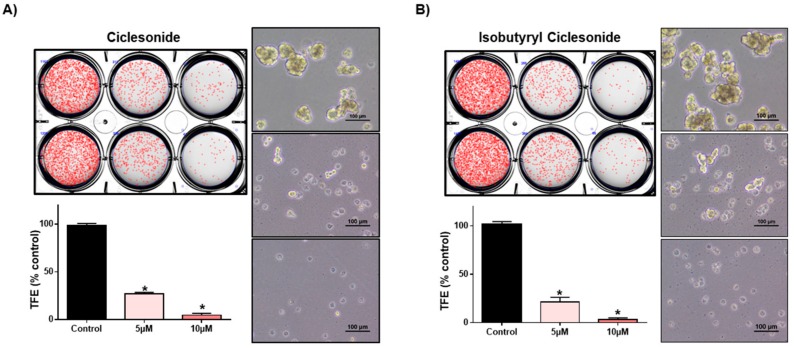
Ciclesonide and isobutyryl ciclesonide reduce tumorsphere formation in lung cancer cells. (**A**,**B**) Tumorspheres derived from A549 cells were cultured for 7 days with cancer stem cell (CSC) medium. Treatment with ciclesonide (5 and 10 μM) reduced tumorsphere formation to 5%. The data of triplicate experiments are shown as the mean ± SD. Compared with control, * *p* < 0.05. (**C**) A549 cells were cultured in a 96-well plate with prednisone and dexamethasone. The growth of cancer cells was measured with MTS reagents. (**D**) Treatment with prednisone and dexamethasone (40 and 80 μM) did not reduce tumorsphere formation efficiency (TFE). The data of triplicate experiments are shown as the mean ± SD. Compared with control, * *p* < 0.05.

**Figure 4 ijms-21-01014-f004:**
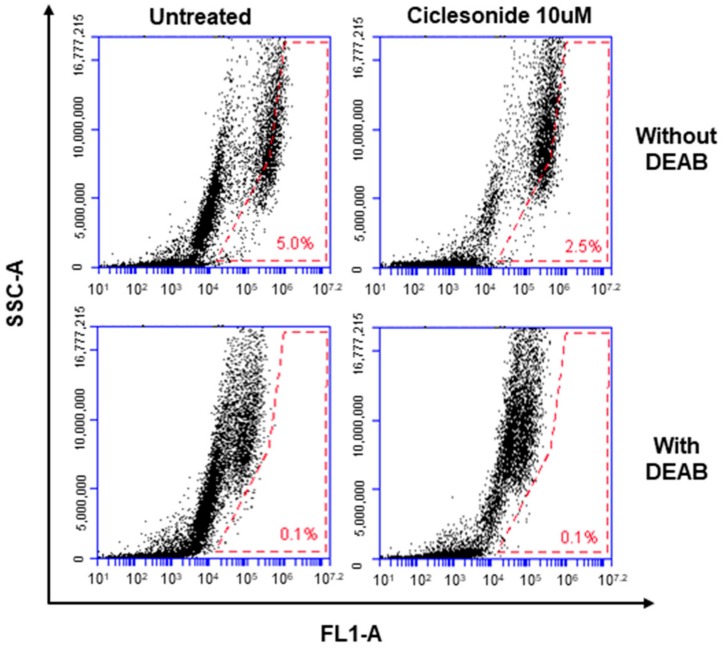
ALDH assay. A549 cancers were incubated in a 6-well plate with ciclesonide. The ALDH activity of lung cancer was assayed using an ADELFLUOR assay kit. The portion of ALDH1-positive cells was quantified by the ADELFLUOR assay. The ALDH-positive fraction was reduced after ciclesonide treatment. The lower parts show representative red-dot plots of the negative control experiment after treatment with the ALDH inhibitor (DEAB). The upper parts indicate ALDH-positive cells that were not treated with DEAB. Values in the red-dot plots show the percentage of ALDH-positive cells.

**Figure 5 ijms-21-01014-f005:**
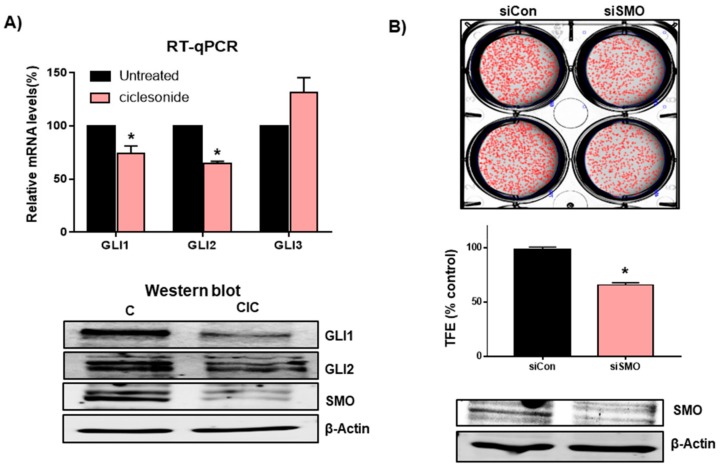
Ciclesonide blocks the Hedgehog signaling pathway through downregulation of the GLI1, GLI2, and Smoothened (SMO) proteins. (**A**) Ciclesonide treatment (10 μM) decreased Hedgehog signaling-related gene expression in the A549-derived tumorspheres. The mRNA levels of GLI1, GLI2, and GLI3 in tumorspheres were examined using specific primers after treatment with ciclesonide. β-actin was used as a loading control. The data of triplicate experiments are shown as the mean ± SD. Compared with control, * *p* < 0.05. The protein levels of GLI1, GLI2, and SMO were analyzed in ciclesonide-treated tumorspheres using specific anti-GLI1, anti-GLI2, and anti-SMO antibodies for immune-blotting. β-actin was used as a loading control. (**B**) Effect of SMO on tumorsphere formation using SMO small interfering RNA (siRNA). Tumorspheres of siRNA-transfected cells were incubated for 7 days with complete CSC medium, and tumor formation efficiency (TFE) was assayed. Representative images of Western blots are shown. The data of triplicate experiments are shown as the mean ± SD. Compared with control, * *p* < 0.05.

**Figure 6 ijms-21-01014-f006:**
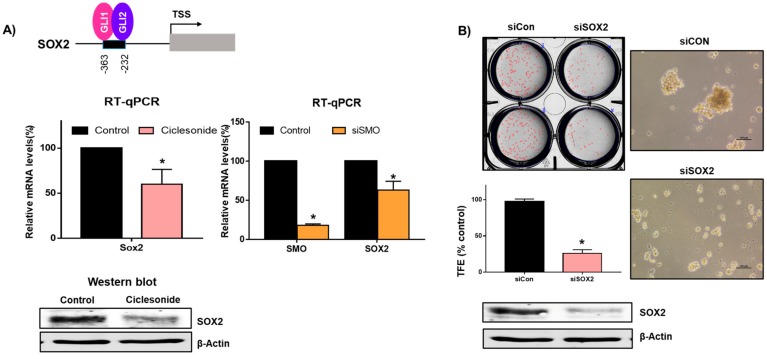
The effect of ciclesonide on SOX2 expression and tumorsphere formation. (**A**) Treatment of the tumorsphere for 48 h with ciclesonide (10 μM) decreased the mRNA and protein levels of the *SOX2* gene. The SMO gene was knocked-down by using siRNA of the SMO gene. The transcripts of the SOX2 and SMO genes were assayed with specific real-time RT-qPCR primers. The SOX2 protein was assayed with an anti-SOX2 antibody. β-actin was used as an internal control. (**B**) siRNA-induced silencing of SOX2 reduced the expression of SOX2 and the formation of tumorspheres. Tumorspheres of siRNA-treated cells were incubated for 1 week with complete CSC medium, and tumorsphere formation efficiency (TFE) was assayed. Representative images of Western blots are shown. The data of triplicate experiments are shown as the mean ± SD. Compared with control, * *p* < 0.05.

**Figure 7 ijms-21-01014-f007:**
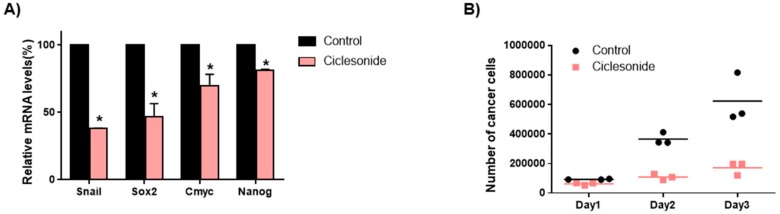
The effect of the anti-asthma medicine ciclesonide on CSC traits in lung cancer. (**A**) The mRNA levels of Snail, SOX2, Nanog, and c-Myc in tumorspheres were analyzed using specific primers after treatment with ciclesonide. β-actin was used as a loading control. The data of triplicate experiments are shown as the mean ± SD. Compared with control. * *p* < 0.05. (**B**) Ciclesonide prevented tumor growth. After treatment with ciclesonide, tumorspheres were divided into single cells, and equal numbers of cells were plated in a 6 well plate. One day after plating, the cells were calculated. After 2 or 3 days, the cells were calculated in triplicate, and the mean value was plotted. Compared with control, * *p* < 0.05.

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
