# Peer review of "The FDA-Approved Anti-Asthma Medicine Ciclesonide Inhibits Lung Cancer Stem Cells through Hedgehog Signaling-Mediated SOX2 Regulation"

_ijms, 2020, doi:10.3390/ijms21031014_

Round 1

Reviewer 1 Report

In this manuscript, the authors showed that ciclesonide, a glucocorticoid, inhibits the proliferation of lung cancer cells and CSC formation; however, similar glucocorticoids, dexamethasone and prednisone, do not inhibit CSC formation. The authors also showed that ciclesonide inhibits Hedgehog signaling pathway, and also the expression of SOX2 and other reprogramming factors for stem cells. From these results, the authors suggest that ciclesonide inhibits lung CSC formation through suppression of Hedgehog signaling and SOX2.

I think that the authors’ results are potentially interesting, especially for future cancer treatment. However, I think that there are several unresolved concerns. Ciclesonide suppressed all phenomena analyzed in this manuscript, such as cell proliferation, hedgehog signal, the expression of Sox2 and other reprogramming factors. However, it is not clear whether these inhibitions are specific for hedgehog signal and rhe induction of SOX2 or caused by an inhibitory effect on overall cellular activity, such as metabolic inhibition, because the authors did not show control phenomenon that not be inhibited by ciclesonide. I think they should show this.

In addition, in Figure 5B, the authors showed that inhibition of SMO expression caused tumor sphere formation of lung cancer cells. If clesonide suppresses CSC formation through inhibition of hedgehog signal, it is considered that the SOX2 expression is suppressed by inhibition of SMO expression, SMO inhibitor or autocrine Shh neutralizing antibody. I think that the authors should show these experiments, to consolidate their model.

Reviewer 2 Report

The authors Choi et al make a compelling case for drug repurposing of an FDA-approved anti-asthma medicine (ciclesonide) for its potential use in lung cancer treatment. The paper describes the inhibitory effect of ciclesonide on lung cancer and cancer stem cells. The authors go further to propose a mechanistic explanation for the observed activity, suggesting that ciclesonide is involved in the suppression of the Hedgehog signalling, pathway known to be activated in solid tumours.

The paper is timely and highlights the potential benefits of drug repurposing.

The study is well designed, however there are still a number of  issues that need to be addressed.

What is the significance of the ciclesonide concentration used in the experiments. Is  it comparable to the current recommended levels of drugs used in asthma treatment? Are they above or bellow the recommended intake  While the presented results clearly indicate a statistical significance reduction in gene expression levels of GL1 and GL2 in the A549 cancer cell line, how does the resulting expression value compares with the level of transcription in normal healthy controls. In the absence of normal lung cell line data, the authors might attempt to answer this it might be worth to examine the FC ratio between cancer and healthy controls (e.g. use TCGA data from XenaBrowser) vs FC ratio between treated and untreated samples.  In section 2.2. when the tumour growth is assessed under the ciclesonide treatment, are the mice subjected to the drug from the start? Is there a way to assess the reduction of the tumour size when the treatment is started for cancer at different stages?

Round 2

Reviewer 1 Report

I think that this manuscript is now suitable for publication in IJMS.

Author Response

Thank you so much for Reviewer1 comments.

Reviewer 2 Report

I appreciate the authors attempts to address the issues raised, however a couple still remain unclear:

Issue 1:

Regarding the concentration of the dexamethasone, in the paper of Brady et al 1987, the authors indeed report a dosage of 40-200mg per patient. However it is worth noting that the average weight of patients reported in that study is 60kg. In the paper of Choi et al, the authors indicate that they use 135-270mg/50kg patient that is 162-324mg/ 60kg patient, that is almost 1.5 times higher that the dosage reported in the 1987 paper, therefore I would like the authors to comment on how this dosage might affect the results observed.  The concentration of ciclesonide reported as 500mg / 50kg  person is extremely high, given the prescribed Alvesco dosage is in the range of 160-(320x2)μg for an adult person. Also 50kg selected as weight of patient is extremely low as the average weight reported in patient studies is around 60kg. This would suggest that the concentration of ciclesonide used in the experiment is extremely high 600mg vs 600μg. The authors should comment on the choice of concentration and if any toxicity effects should be considered.

The authors should clearly comment in the manuscript on the used concentration levels and their biological significance. 

Issue 2:

The authors show in Response letter Figure 1 the relative values for the gene expression in lung adenocarcinoma vs normal, from TCGA data. However they should have shown if there is any statistical significant change in the expression levels of the genes of interest. Also, to calculate the fold change, the authors should have calculated the ratio between the mean expression in cancer vs the mean expression in normal tissue, they could have calculate the same ratio in their experiments and compared the two, this comparison would give a clearer  idea of how significant is the reduction in gene expression levels of GL1 and GL2 in the A549 cancer cell line.

Issue 3: - addressed 

Author Response

We submit Reviewer 2 comments.

Round 3

Reviewer 2 Report

I am satisfied with the answers received.